# An Overview of Vanadium and Cell Signaling in Potential Cancer Treatments

**Valeria Alejandra Ferretti [1] and Ignacio Esteban León [1,2,]*** 

[1] Cequinor (UNLP, CCT-CONICET La Plata, Asociado a CIC), Departamento de Química, Facultad de Ciencias Exactas, Universidad Nacional de La Plata, Blvd. 120 N° 1465, La Plata 1900, Argentina; vferretti@med.unlp.edu.ar

[2] Cátedra de Fisiopatología, Departamento de Ciencias Biológicas, Facultad de Ciencias Exactas, Universidad Nacional de La Plata, 47 y 115, La Plata 1900, Argentina

[*] Correspondence: ileon@biol.unlp.edu.ar

**Abstract:** Vanadium is an ultratrace element present in higher plants, animals, algae, and bacteria. In recent years, vanadium complexes have been studied to be considered as a representative of a new class of nonplatinum metal anticancer drugs. Nevertheless, the study of cell signaling pathways related to vanadium compounds has scarcely been reported on and reviewed thus far; this information is highly critical for identifying novel targets that play a key role in the anticancer activity of these compounds. Here, we perform a review of the activity of vanadium compounds over cell signaling pathways on cancer cells and of the underlying mechanisms, thereby providing insight into the role of these proteins as potential new molecular targets of vanadium complexes.

**Keywords:** vanadium biochemistry; cell signaling; cancer; anticancer agents



## 1. Introduction

Metallodrugs have a wide field of therapeutical activities towards several pathologies, including infections, neurodegenerative diseases, diabetes, and cancer [1–4]. Platinum compounds, in special cisplatin (CDDP), carboplatin, and oxaliplatin, are the most relevant and effective metallodrugs [5]. Nevertheless, the lack of specificity, poor absorption, and chemoresistance limit the current use in the clinic. Therefore, medicinal inorganic chemistry focuses on the design and synthesis of novel metal-based drugs aiming to overcome these complications [6,7]. In this sense, the anticancer activity of vanadium complexes has been extensively in vitro and in vivo tested on several types of cancer cell lines. VO (oxidovanadium) flavonoids are an important group of compounds with selective antitumor effects on bone cancer cells [8–10]. Metvan (V$^{IV}$O(SO$_4$)(4,7-Mephen)$_2$) is a vanadium complex with anticancer activity on different human tumor cell lines, such as leukemia cells, multiple myeloma cells, and solid tumor cells (brain, prostate, breast, ovarian, etc.) [11–14]. Moreover, vanadocene derivatives have shown important anticancer effects on human cancer cell lines, mainly involving liver and testicular tumors. Another interesting group with antitumor properties is vanadium compounds with heterocycles and Schiff bases. Their complexes have shown antitumor actions on bone, breast, and colon cancer cells [15–17].

The role of vanadium in the regulation of cell signaling pathways converts it into a prospective therapeutic agent to be employed in various pathologies. However, the known activation pathways targeted by vanadium compounds are narrowly reported in the literature and, thus far, these data for the discovery of novel intracellular targets in cancer have not been widely analyzed.

In this review, we present an outline of the cell pathways activated or inactivated by vanadium complexes on cancer cells and the relationship with the anticancer activity of these compounds. This overview is expected to achieve a better understanding of

these intracellular signaling pathways and, thereby, may facilitate the design of vanadium complexes with promising therapeutic applications as well as the comprehension of side effects derived from the use of the vanadium compounds as therapeutic agents.

## 2. Vanadium in Cancer Therapeutics

Vanadium is a transition metal that exists in different oxidation states ranging from −1 to +5. At pharmacological doses, compounds with vanadium III, IV, and V show biologically significant effects such as insulin imitators [18,19], growth factor-like activity [20], and antitumoral properties [9,21–24].

Various effects of vanadium derivatives have been observed on the activity of several enzymes, especially those related to phosphate reactions. Vanadium inhibits various AT-Pases with different effectiveness [25]. Furthermore, vanadium compounds also inhibit several phosphatases, such as alkaline phosphatase, acid phosphatase, and tyrosine–protein phosphatases (PTPases) [26]. The PTPases activate or inhibit intracellular signaling pathways, triggering different biological events in a cascade manner, among these being cancer signaling pathways.

## 3. Cancer-Related Signaling Pathways Activated by Vanadium Complexes

Diverse assays performed in vitro or in vivo demonstrated that vanadium compounds can activate different cancer signaling pathways, and so exert their antitumoral action.

### 3.1. MAPK (Mitogen-Activated Protein Kinases)/ERK (Extracellular Signal-Regulated Kinase) Signaling Pathway

The MAPK/ERK pathway is one of the early signaling pathways for cell cycle progression [27,28]. An essential role in cancer development is attributable to alterations regarding different molecular pathways such as the MAPK involved in regulating cell growth. The uncontrolled activation of MAPKs is due to diverse gene mutations, some of which regulate the constitutive activation of the B-Raf protein kinase (cytoplasmic protein) that induces the activation of the mitogen-activated protein kinase (MEK), which in turn activates the extracellular signal-regulated kinase (ERK), the final effector of the pathway, inducing the transcription of target genes that generate the cell entering the cell cycle.

Bis(acetylacetonate)-oxidovanadium(IV) (VO(acac)$_2$) and sodium metavanadate (NaVO$_3$), two well-known antidiabetic compounds, have shown an antiproliferative effect through inducing a G$_2$/M cell cycle arrest and an elevation in reactive oxygen species (ROS) levels in human pancreatic cancer cell line AsPC-1. It is important to highlight that NaVO$_3$ converts to H$_2$VO$_4^-$ at physiological conditions, in which the cellular assays were carried out.

ROS are fundamental agents in cell fate. Their intracellular accumulation in normal cells means the oxidation of cellular components, such as nucleic acids, proteins, and lipids. These oxidative reactions cause extensive damage and in cases of irreparable damages, they stimulate apoptosis [29]. In this sense, it has been found that these two vanadium compounds prompt the activation of the MAPK/ERK signaling pathway in a dose- and time-dependent manner. Both compounds generate an increase in the phosphorylated ERK levels; therefore, these vanadium compounds could cause a cell cycle arrest and a high elevation in the ROS levels by positively modulating the MAPK/ERK signaling pathway and, thus, causing tumor suppression [30].

Another study revealed an antiproliferative activity of the inorganic anion vanadate(V) (VN) and the oxidovanadium (IV) complex (VO(1,2-dimethyl-3-hydroxy-4(1H)-pyridinonate)$_2$) (VS2) on the melanoma A375 cell line. The authors demonstrated that both vanadium (IV, V) species displayed an antitumoral activity by arresting the cell cycle and causing apoptosis across intracellular ROS production, ERK, and retinoblastoma protein (Rb) dephosphorylation and p21Cip1 overexpression [31]. The retinoblastoma protein (Rb) constitutes an essential control point for the switch from the G$_1$ phase to the S phase, whereas the cyclin/CDK complex inhibitor p21Cip1 is involved in the cell cycle blockade.

Vanadium compounds are characterized by their ability to regulate stem cell differentiation [24]. Recently, N,N-bis(salicylidene)-o-phenylenediamine vanadium(IV) oxide was reported to upmodulate osteoblast differentiation. Thus, the V(IV) species seem to stimulate the differentiation and mineralization of the mesenchymal stem cells via the activation of the ERK signaling pathway and the subsequent improvement of the NF-κB (nuclear factor kappa-light-chain-enhancer of activated B cells) mediated action. Furthermore, it has been established that ERK is involved in the increase in the transcriptional activity of NF-κB. Thus, V(IV) may modulate both ERK and NF-κB pathways, and both pathways would act jointly to encourage osteoblasts [32].

In summary, different research suggests that some vanadium compounds can affect the MAPK/ERK signaling pathway, prompting a cell cycle arrest, an increase in the apoptosis, and, thus, causing tumor reduction.

### 3.2. PI3K (Phosphatidylinositol 3-Kinase)/AKT (Protein Kinase B) Signaling Pathway

Oncogenic RET/PTC1 (receptor tyrosine kinase/type two C phosphatase 1) chromosomal rearrangements are hallmarks of thyroid papillary carcinoma. The resulting protein, mainly, through tyrosine 451, is responsible for the activation of pathways controlling cell survival, including the PI3K/Akt cascade. The PI3K/Akt signaling cascade has an important role in the control of cell survival, metabolism, and motility, with unsuitable signals through this pathway occurring habitually in cancer [33]. Vanadium compounds were revealed to have antitumoral potential in thyroid papillary carcinoma, among others [34]. In this sense, a study realized utilizing papillary thyroid carcinoma-derived TPC-1 cells revealed that a low dose of orthovanadate (OV) (100 nM) induces a pro-proliferative response. In contrast, treatment with inhibitory amounts of the compound (10 μM) resulted in a greater phosphorylation of tyrosine 451 of RET/PTC1, triggering the mTOR/S6R branch of the PI3K/Akt signaling pathway. These concentrations of the drug also generate typical features of apoptosis, including DNA fragmentation, the loss of mitochondrial membrane potential, production of ROS, and activation of caspase-3 [35].

Another study realized utilizing MCF7 human breast cancer epithelial and A549 lung adenocarcinoma cells revealed that vanadium produces a significant decline in cancer cell viability, decreasing H-ras signaling and metalloproteinase-2 (MMP-2) expression by raising ROS-mediated apoptosis [36]. On the other hand, it is well known that vanadium compounds are effective in diabetes treatment due to their insulin-mimetic behavior and the stimulation of glucose catchment [21,37]. Pandey et al. [38] demonstrated that vanadyl sulfate stimulates the ras-ERK pathway through the activation of PI3K, and they presumed that the stimulation of the PI3-K/ras/ERK cascade plays an essential role in mediating the insulin-mimetic effects of the vanadium compounds (Figure 1).

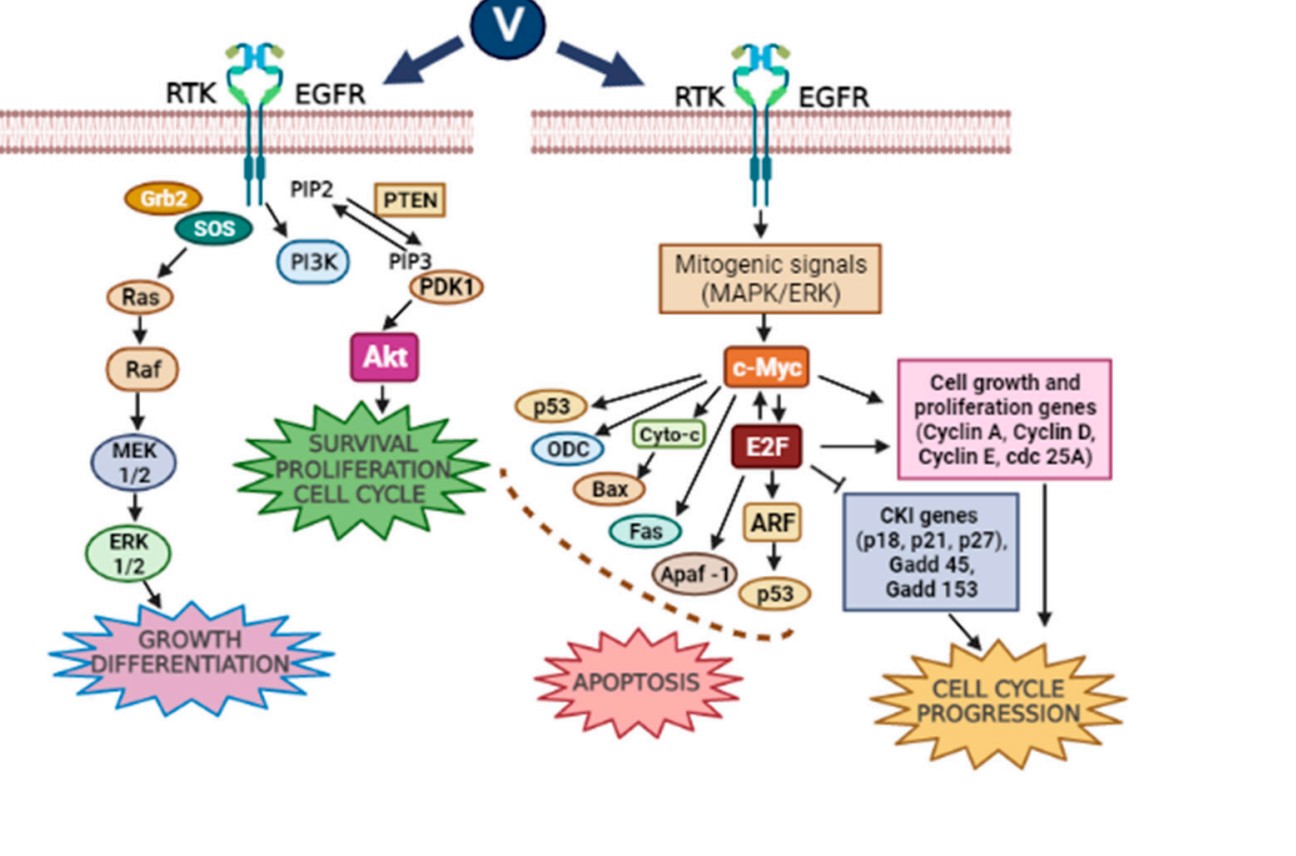

**Figure 1.** MAPK/ERK cell signaling pathways affected by vanadium compounds. Adapted from [39].

In consequence, it was observed that the vanadium compounds can lead to an increase in the ROS-mediated apoptosis and a decline in cancer cell viability by stimulating the PI3K/Akt signaling pathway.

### 3.3. Caspase Signaling Pathway

Vanadium is characterized by its capacity to stimulate the apoptotic machinery in cancer cells through the upregulation of the primary apoptosis proteins. In relation to this, we found that the oxidovanadium(IV) flavonoids caused a cell cycle arrest and activated caspase-3, triggering apoptosis in a human osteosarcoma cell line MG-63 [8,15]. In another report, we studied the mechanism of action of the oxidovanadium(IV) complexes with the flavonoids silibinin $Na_2[VO(silibinin)_2]\cdot6H_2O$ (VOsil) and chrysin $[VO(chrysin)_2EtOH]_2$ (VOchrys), utilizing human colon adenocarcinoma-derived cell line HT-29. In this work, we found that the VOchrys caused a cell cycle arrest in the $G_2/M$ phase, while VOsil activated caspase-3, triggering the cells directly into apoptosis [40] (Figure 2). Moreover, VOsil diminished the NF-κB activation via increasing the sensitivity of cells to apoptosis [37]. As mentioned before, orthovanadate also generates an activation of caspase-3 in papillary thyroid carcinoma cells [35]. Moreover, it has been demonstrated that vanadium compounds induce apoptosis and the expression of caspase-3, Bcl-2, and Bax, which regulate cell apoptosis in neuronal cells [41,42].

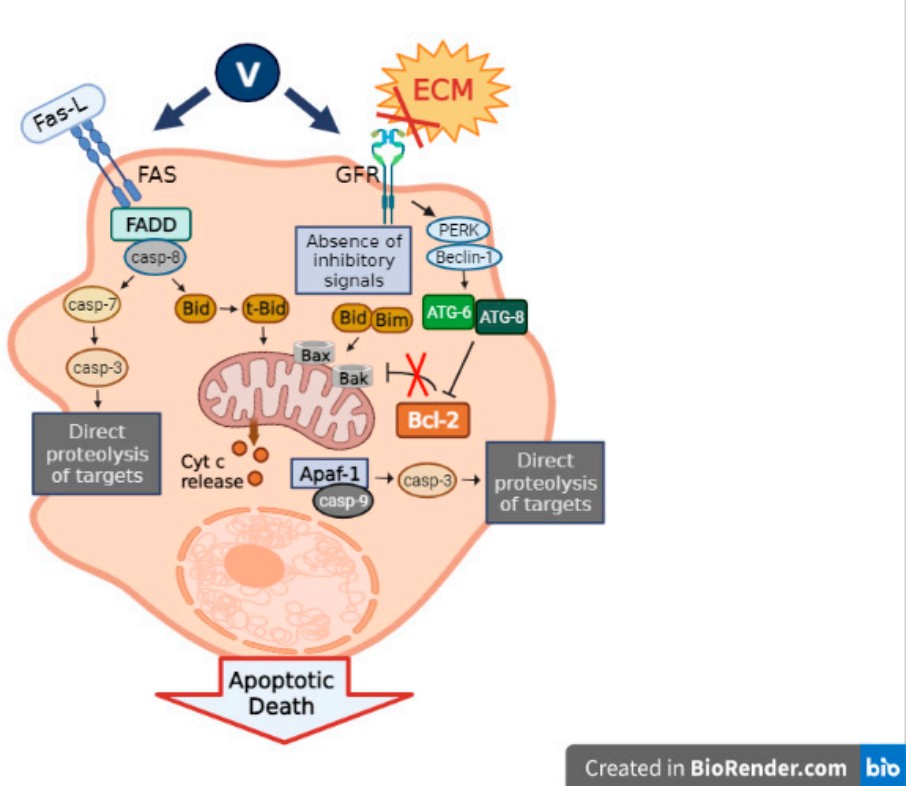

**Figure 2.** Apoptosis cell pathways activated by vanadium compounds. Adapted from [43].

Briefly, several vanadium compounds can exert their effects on the caspase signaling pathway, generating a rise in cancer cell apoptosis and, thus, tumor suppression.

### 3.4. JAK (Janus Kinase Protein)/STAT (Signal Transducer and Activator of Transcription Proteins) Signaling Pathway

Vanadium is also considered an air contaminant released into the atmosphere by burning fossil fuels. Moreover, its carcinogenic potential has been assessed to establish permissible limits of exposure at workplaces. Gonzalez—Villalva et al. [44,45] detected a growth in the number and size of platelets and their precursor cells and megakaryocytes in bone marrow and spleen as a consequence of the vanadium exposure. In another study, performed on mice exposed to vanadium pentoxide, they found an increase in JAK2 ph (phosphorylated Janus kinase 2 protein) and STAT3 ph (phosphorylated STAT3), but a decline in the Mpl (myeloproliferative leukemia protein) receptor [45]. In consequence, they concluded that the vanadium pentoxide could activate the JAK/STAT pathway and decrease the Mpl receptor; thus, leading to a condition analogous to essential thrombocythemia. They also proposed that the reduction in Mpl was a negative feedback mechanism after the JAK/STAT activation.

The Mpl receptor does not have tyrosine kinase activity, but is constitutively linked to JAK2, which has tyrosine kinase activity that phosphorylates the signal transducer and activates the transcription of STAT3. STAT3 ph translocates to the nucleus to function as a transcription factor, which activates genes that stimulate survival and apoptosis inhibition, such as Bcl-xl, p27, p21, and cyclin D1. Since megakaryocytes are platelet precursors, their modification affects platelet morphology and function, which might have consequences in hemostasis; therefore, it is imperative to continue assessing the effects of chemicals and contaminants on megakaryocytes and platelets.

Apparently, some vanadium compounds in certain doses can activate the JAK/STAT signaling pathway and, hence, prompt an increase in the number and size of the platelets, a condition analogous to essential thrombocythemia.

*3.5. Nrf2 (Nuclear Factor Erythroid 2-Related Factor 2)/HO-1 (Heme Oxygenase-1) Signaling Pathway*

CDDP is one of the first-line anticancer treatments; however, the main limitation of CDDP therapy is the development of nephrotoxicity (25–35% cases), whose specific mechanism primarily includes oxidative stress, inflammation, and cell death. Thus, looking for a potential chemo-protectant, Basu et al. [46] assessed an organo–vanadium complex, vanadium(III)-L-cysteine (VC-III), against CDDP-induced nephropathy in mice.

The VC-III treatment significantly avoided the CDDP-induced generation of ROS, reactive nitrogen species, and the beginning of lipid peroxidation in kidney tissues of the experimental mice. Furthermore, VC-III also extensively returned CDDP-induced depleted activities of the renal antioxidant enzymes, such as superoxide dismutase, catalase, glutathione peroxidase, glutathione- S-transferase, and glutathione (reduced) levels. In addition, the VC-III treatment also quite successfully reduced the expression of proinflammatory mediators, such as NFκβ, COX-2, and IL-6, and activated the Nrf2-mediated antioxidant defense system through the promotion of downstream antioxidant enzymes (HO-1). The treatment with VC-III considerably improved CDDP-mediated cytotoxicity in MCF-7 and NCI-H520 human cancer cell lines. Therefore, VC-III could function as a proper chemo-protectant via stimulating the Nrf2/HO-1 signaling pathway and enhancing the therapeutic window of CDDP in cancer patients (Figure 3).

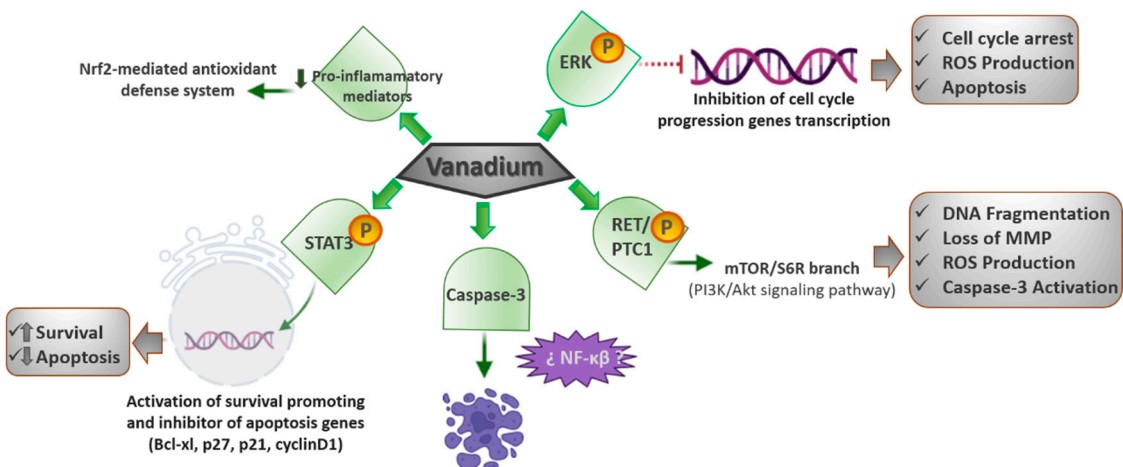

**Figure 3.** Upregulated proteins involved in survival and cell death induced by vanadium complexes.

## 4. Cancer-Related Signaling Pathways Inactivated by Vanadium Complexes

Interestingly, many works have revealed that vanadium compounds are also capable of inhibiting several cancer signaling pathways, and so work as antitumor and antimetastatic agents.

*4.1. FAK Signaling Pathway*

Recently, we demonstrated that oxidovanadium (IV)–chrysin and oxidovanadium (IV)–clioquinol (VO(CQ)$_2$) complexes prevent the activation of focal adhesion kinase (FAK), reducing the cell proliferation in human osteosarcoma cells [10,47]. Our results showed that VO(CQ)$_2$ is located near the activation loop of the kinase domain and establishes interactions with residues in the ATP binding site. Particularly, vanadium–clioquinol exhibited a dual behavior at 2.5 μM, since the Tyr576 and Tyr577 sites were upmodulated; however, at 10 μM, the phosphorylation of Tyr576 and Tyr577 declined 14-fold [48]. Likewise, in another work, we reported that the oxidovanadium (IV)–chrysin complex ([VO(chrysin)$_2$EtOH]$_{2)}$ upmodulated the Tyr577 site of phosphorylation, but downmodulated Tyr397 [10]. The Tyr397 site of tyrosine phosphorylation is the most active and common site for the autocatalytic function of FAK [49]. Thus, these results would indicate that [VO(chrysin)$_2$EtOH]$_2$

selectively repressed the autophosphorylation activity of FAK kinase, directly affecting the Tyr397 site.

On the other hand, FAK is a tyrosine kinase that carries out a crucial role in the adhesion, survival, motility, angiogenesis, and metastasis of cancer cells [50]. Moreover, FAK is overexpressed in numerous kinds of solid and nonsolid tumors [51], so FAK has been suggested as a therapeutic target [52].

Furthermore, vanadium compounds diminished the cell migration in 2D and 3D human bone cancer cell models. Additionally, VO(CQ)$_2$ considerably reduced the activity of MMP-2 and MMP-9 in a dose-dependent way, advising the direct relationship between FAK inhibition and the inactivation of MMPs (matrix metalloproteinases).

### 4.2. Autophagy Signaling Pathway

Recent studies have demonstrated the great ability of vanadium to regulate the process of autophagy. Inorganic sodium orthovanadate (SOV) converts to an active agent at physiological conditions in H$_2$VO$_4$$^-$, potentially prompting cell apoptosis and the prevention of autophagy in human hepatocellular carcinoma (HCC) cells, in vitro and in vivo, concurrently. Additionally, a further decrease in autophagy by 3-methyladenine (3MA) considerably improves SOV-induced apoptosis in HCC cells, while rapamycin could reverse such autophagy inhibition and decrease the apoptosis-stimulating effect of SOV in HCC cells, both in vitro and in vivo. The results showed that such an autophagy inhibition effect plays a pro-death role [53]. Likewise, nano-sized paramontroseite VO2 nanocrystals (P-VO2) prompted cyto-protective, rather than death-inducing, autophagy in cultured HeLa cells. P-VO2 also prompted the upmodulation of hemeoxygenase-1 (HO-1), a cellular protein with a proved role in protecting cells against death under stress conditions. The autophagy inhibitor 3-methyladenine considerably repressed HO-1 upregulation and augmented the rate of cell death in cells treated with P-VO2, while the HO-1 inhibitor protoporphyrin IX Zn(II) (ZnPP) improved the existence of cell death in the P-VO2-treated cells, while displaying no effect on the autophagic response induced by P-VO2. Likewise, Y$_2$O$_3$ nanocrystals, a control nanomaterial, prompted death-inducing autophagy without affecting the level of expression of HO-1 and the pro-death effect of the autophagy prompted by Y$_2$O$_3$. These data represent the first report on a novel nanomaterial-induced cyto-protective autophagy, possibly through the upregulation of HO-1, and potentially leading to new opportunities for taking advantage of nanomaterial-induced autophagy for cancer therapeutic applications [54].

### 4.3. TGFβ (Transforming Growth Factor-β)- EMT (Epithelial to Mesenchymal Transition) Signaling Pathway

The EMT plays a crucial role in tumor advancement and metastasis as an essential event for cancer cells to generate the metastatic niche. TGF-β has been revealed to play a significant role as an EMT inducer in several stages of carcinogenesis. Some studies have revealed that vanadium inhibits the metastatic potential of tumor cells by decreasing MMP-2 expression and prompting ROS-dependent apoptosis. Petanidis et al. [55] described, for the first time, the inhibitory effects of vanadium on (TGF-β)-mediated EMT, followed by the downmodulation of cancer stem cell markers in human lung cancer adenocarcinoma A549 and breast cancer MDA-MB-231 epithelial cell models. The results showed a blockage of TGF-β-mediated EMT by vanadium and a decay in the mitochondrial potential of cancer cells related to EMT and cancer metabolism. Moreover, they reported that the combination of vanadium and carboplatin results in a G$_0$/G$_1$ cell cycle arrest and the sensitization of cancer cells to carboplatin-induced apoptosis. This knowledge could be valuable for targeting the cancer cell metabolism and cancer stem cell-mediated metastasis in aggressive chemoresistant tumors.

*4.4. Notch-1 Signaling Pathway*

In recent work, it was found that vanadium compounds can suppress the growth of the MDA-MB-231 cell line, a model of the most aggressive and therapy-resistant triple-negative breast cancer. The [VO(bpy)$_2$Cl]Cl complex (bpy = bipyridyl) generated a rise in caspase-3 levels and, thus, an induction of the apoptotic cell death [56]. Moreover, the authors found a decrease in the Notch1 gene expression, thereby inhibiting the Notch-1 pathway. The Notch signaling pathway is a greatly conserved cell signaling system, which plays a major role in the regulation of embryonic development and is dysregulated in numerous types of cancers, such as lung and breast cancers [57–59]. Additionally, the inactivation of Notch signaling has been demonstrated to have antiproliferative effects on T-cell acute lymphoblastic leukemia in cultured cells and a mouse model [59] (Figure 4).

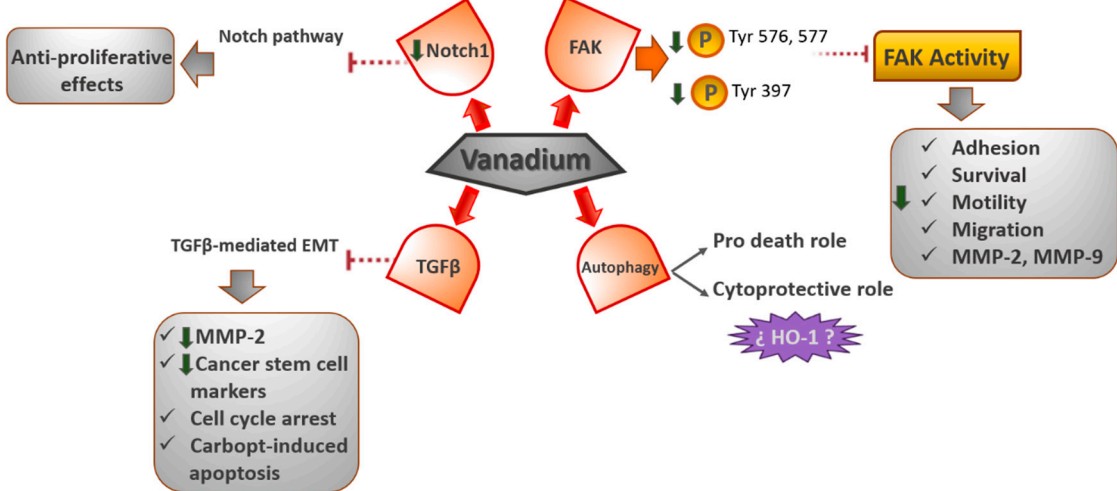

**Figure 4.** Downregulated proteins involved in survival and cell death induced by vanadium complexes.

**5. Conclusions**

Over recent years, vanadium compounds have been studied and considered as representatives of a new class of nonplatinum metal antitumor agents. Nevertheless, knowledge surrounding cell signaling pathways related to vanadium drugs is scarce. In this review, we presented a brief overview of the cell pathways activated or inactivated by vanadium complexes on cancer cells and the relationship with the anticancer activity of these compounds. The MAPK/Erk, PI3K/Akt, and caspase family and JAK/Stat signaling pathways were stimulated by the vanadium compounds, prompting a cell cycle arrest, ROS production, and apoptosis towards different types of cancer cells. The Nrf-2 was also activated by the vanadium; however, in this case, it seemed to enhance the defense system and functioned as a chemoprotective. On the other hand, the vanadium complexes would also stimulate the JAK/Stat signaling pathway, generating a growth effect in platelets and their precursors.

Likewise, the FAK, TGF-B/EMT, Notch-1, and autophagy signaling pathways were inactivated by vanadium compounds, potentially leading to an increase in cell cycle arrest and apoptosis, and a decrease in cellular migration and adhesion, generating tumor suppressor effects.

Taken together, hopefully this review will generate further understanding of the activity of vanadium compounds over cell signaling pathways on cancer cells and of the underlying mechanisms, and may thereby facilitate the design of vanadium complexes with promising therapeutic applications to improved cancer treatments.

**Author Contributions:** Conceptualization, V.A.F. and I.E.L. investigation, V.A.F. and I.E.L.; writing—original draft preparation, V.A.F.; writing—review and editing, I.E.L.; supervision, I.E.L.; funding acquisition, I.E.L. All authors have read and agreed to the published version of the manuscript.

**Funding:** This work was partly supported by UNLP (PPID X053) CONICET (PIP 2051), and ANPCyT (PICT 2019-2322) from Argentina.

**Acknowledgments:** The authors thank Consejo Nacional de Investigaciones Científicas y Técnicas (CONICET) since I.E.L. is a member of the Researcher Career.

**Conflicts of Interest:** The authors declare no conflict of interest.

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
