# Peer review of "An Overview of Vanadium and Cell Signaling in Potential Cancer Treatments"

_inorganics, doi:10.3390/inorganics10040047_

Round 1

Reviewer 1 Report

Comments to Authors
Vanadium is becoming increasingly important in the field of health and this review gives a good overview of the use of this element as an anticancer agent and the signaling pathways it has been shown to influence.
1. Title: The review is focused on the antitumor activity of vanadium, but I consider that the title is ambiguous in this regard, because it is understood as the role of vanadium as a carcinogenic agent. I suggest refining the title, as well as adding the word "Review".
2. Introduction: The abbreviation "VO-flavonoids" is not described in the text.
3. Section 3: Cancer-related signalling [...]: At the end of each sub-section I suggest that you add a short general explanation of the implications that vanadium would have on each pathway, so that the reader understands specifically: what does it imply that vanadium affects that signalling pathway?
4.    Sub-section 3.4. JAK-STAT the reference of Gonzalez-Villalba et al. should read Gonzalez-Villalva et al. as written in the references.
5.    It is important that the authors write the "Conclusion" of the review.
6.    Review the format of the references: references 8, 9, 10, 46 and 47 have only the authors' initials. Also, if the article is in an electronic journal, it is not necessary to put [internet] this format does not apply to this type of references. I suggest reviewing the style of the references. 

Author Response

Comments to Authors

Vanadium is becoming increasingly important in the field of health and this review gives a good overview of the use of this element as an anticancer agent and the signaling pathways it has been shown to influence.

We specially thank to the reviewer 1 for the constructive comments.

  1. Title: The review is focused on the antitumor activity of vanadium, but I consider that the title is ambiguous in this regard, because it is understood as the role of vanadium as a carcinogenic agent. I suggest refining the title, as well as adding the word "Review".

According to the reviewer´s comments we have been changed the title in the new version of the manuscript.

  1. Introduction: The abbreviation "VO-flavonoids" is not described in the text.

Done.

  1. Section 3: Cancer-related signalling [...]: At the end of each sub-section I suggest that you add a short general explanation of the implications that vanadium would have on each pathway, so that the reader understands specifically: what does it imply that vanadium affects that signalling pathway?

According to the reviewer´s suggestion we have been added a short explanation about vanadium implication in each pathway.

  1.    Sub-section 3.4. JAK-STAT the reference of Gonzalez-Villalba et al. should read Gonzalez-Villalva et al. as written in the references.

Done.

  1.    It is important that the authors write the "Conclusion" of the review.

According to the reviewer´s comments we have been added a conclusion.

  1.    Review the format of the references: references 8, 9, 10, 46 and 47 have only the authors' initials. Also, if the article is in an electronic journal, it is not necessary to put [internet] this format does not apply to this type of references. I suggest reviewing the style of the references. 

Done.

Reviewer 2 Report

The authors provide a brief overview of the potentiality of vanadium (vanadate and complex vanadium compounds) in the treatment of diverse forms of cancer. The article is of general interest, and I recommend publication, subject to a couple of clarifications (see comments below), and amendment of the English throughout.

Comments:

1) Abstract: Vanadium is not only present in plants and animals, but also in, e.g., algae and bacteria.

2) p. 2, about centre: The statement “related to phosphate reactions in studies performed in cell-free systems” lacks clarity.

3) p. 3, line 2: NaVO3 does not exist in aqueous media.

4) p. 3, 3rd paragraph: “The vanadium is …” should read “Vanadium compounds are …”?

5) p. 3, end of 3rd paragraph: The last sentence in this paragraph lacks clarity.

6) p. 4, about centre: “glucose consumption” connotes “glucose degradation”?

7) p. 6, CDDP ?

9) p. 7, about centre: The statement “vanadium-cliochinol has exhibited dual behaviour …” lacks clarity.

10) p. 8, line 6: The actually active agent at physiological conditions in H2VO4- (and not sodium metavanadate).

11) p. 8, section 4.2: What is “a pro-death role” supposed to connote?

12) p. 9: [VO (bpy)2 Cl] Cl is [VO(bpy)2Cl]Cl (bpy = bipyridyl)?

Author Response

The authors provide a brief overview of the potentiality of vanadium (vanadate and complex vanadium compounds) in the treatment of diverse forms of cancer. The article is of general interest, and I recommend publication, subject to a couple of clarifications (see comments below), and amendment of the English throughout.

We specially thank to the reviewer 2 for the constructive comments.

Comments:

1) Abstract: Vanadium is not only present in plants and animals, but also in, e.g., algae and bacteria.

According to the reviewer´s comments we have added this information in the abstract section in the new version of manuscript.

2) p. 2, about centre: The statement “related to phosphate reactions in studies performed in cell-free systems” lacks clarity.

According to the reviewer´s comments we have changed this phrase in the new version of manuscript.

3) p. 3, line 2: NaVO3 does not exist in aqueous media.

Done

4) p. 3, 3rd paragraph: “The vanadium is …” should read “Vanadium compounds are …”?

Done

5) p. 3, end of 3rd paragraph: The last sentence in this paragraph lacks clarity.

According to the reviewer´s comments we have changed this sentence in the new version of manuscript.

6) p. 4, about centre: “glucose consumption” connotes “glucose degradation”?

Done

7) p. 6, CDDP ? 

CDDP= cisplatin.

Please, see the section 1 Introduction

9) p. 7, about centre: The statement “vanadium-cliochinol has exhibited dual behaviour …” lacks clarity.

According to the reviewer´s comments we have changed this sentence in the new version of manuscript.

10) p. 8, line 6: The actually active agent at physiological conditions in H2VO4- (and not sodium metavanadate).

Done.

11) p. 8, section 4.2: What is “a pro-death role” supposed to connote?

A pro-death role connotes that inducing and promotes cell death.

12) p. 9: [VO (bpy)2 Cl] Cl is [VO(bpy)2Cl]Cl (bpy = bipyridyl)?  

Done